# Using a Parallel Helical Sensing Cable for the Distributed Measurement of Ground Deformation

**DOI:** 10.3390/s19061297

**Published:** 2019-03-14

**Authors:** Xiushan Wu, Renyuan Tong, Yanjie Wang, Congli Mei, Qing Li

**Affiliations:** 1School of Electrical Engineering, Zhejiang University of Water Resources and Electric Power, Hangzhou 310018, China; wuxiushan@cjlu.edu.cn (X.W.); clmei@126.com (C.M.); 2College of Mechanical and Electrical Engineering, China Jiliang University, Hangzhou 310018, China; wangyanjiexx@163.com (Y.W.); lq_cjlu@163.com (Q.L.)

**Keywords:** geological monitoring, parallel helical sensing cable, time–domain reflectometry (TDR), distributed parameter, characteristic impedance, stretched deformation

## Abstract

Surface and underground stretched deformation is one of the most important physical measurement quantities for geological-disaster monitoring. In this study, a parallel helical sensing cable (PHSC) based on the time–domain reflectometry (TDR) technique is proposed and used to monitor large ground stretched deformation. First, the PHSC structure and manufacturing process are introduced, and then, distributed capacitance, distributed inductance, and characteristic impedance were derived based on the proposed stretched-structure model. Next, the relationship between characteristic impedance and stretched deformation was found, and the principle of distributed deformation measurement based on the TDR technique and PHSC characteristic impedance was analyzed in detail. The function of the stretched deformation and characteristic impedance was obtained by curve fitting based on the theoretically calculated results. A laboratory calibration test was carried out by the designed tensile test platform. The results of multi-point positioning and the amount of stretched deformation are presented by the tensile test platform, multi-point positioning measurement absolute errors were less than 0.01 m, and the amount of stretched deformation measurement absolute errors were less than 3 mm, respectively. The measured results are in good agreement with the theoretically calculated results, which verify the correctness of theoretical derivation and show that a PHSC is very suitable for the distributed measurement of the ground stretched deformation.

## 1. Introduction

In recent years, natural geological disasters, such as earthquakes, landslides, and mudslides, have occurred frequently in earthquake-stricken areas, which are serious threats to human life, safety, and economic security. Ground deformation is one of the most important physical measurement quantities for monitoring and predicting geological disasters. Sensing technology is mainly used to monitor ground deformation, including draw-wire displacement sensor technology [1,2], global positioning system (GPS) technology [3,4], total station observation technique [5], remote sensing technique [6,7,8], synthetic aperture radar (SAR) technology [4,9], laser scanning technology [10], optical fiber sensing technology [11,12,13,14,15], and time–domain reflectometry (TDR) technology [16,17]. 

The draw-wire displacement sensor is a two-point measurement, which can accurately measure the relative displacement between two points, but it is difficult to reflect the whole stretched deformation using this method. GPS technology can monitor three-dimensional displacement information by point distribution measurement [18]. However, the measured position must ensure good satellite signals, and the accuracy of vertical displacement is relatively low. The GPS precise point positioning method based on a 24-h observation period can determine horizontal displacement up to 1.5 cm [19]. The total station instrument is a point distribution measurement. With this method, it is necessary to establish the work point in the place where invisible change is guaranteed [20], and the accuracy is easily affected by temperature, atmosphere, and other factors. Synthetic aperture radar interferometry (InSAR) can extract one-dimensional deformation along the line-of-sight of radar, and SAR image co-registration and pixel offsets can be used to extract two-dimensional deformation along the azimuth and range directions [21]. However, the deformation accuracy is significantly affected by matching window size and oversampling, and terrain effect can cause significant pixel displacements in SAR images over high-relief areas. Laser scanning technology can be used to monitor the three-dimensional displacement of a landslide with high precision by using a non-contact method, but the measurement accuracy is easily affected by the buildings and vegetation on the surface. In [22], two scanning methods were adopted to assess the reasons for premature cracks adjacent to bridge objects. Laser scanning was used to measure geometric surface deformation. and ground penetrating radar inspection was used to assess the road pavement condition. A deformation sensor based on optical fiber can realize the continuous and distributed measurement of micro-deformation without electromagnetic interference. In [23], the experimental results showed that the distributed optical fiber can measure the soil strain during the ground deformation process, and the strain curve responded to the soil compression and tension region clearly. When the amount of deformation is larger than 2 mm, the optical fiber is easily broken due to the limited stretching ability. Then, various special optical fibers can be developed to meet the requirements, including larger deformation and coordination deformation. Time–domain reflectometry (TDR) technology has become a valuable tool for monitoring displacements and locating shear planes in rock–soil slopes. A deformation sensor based on a coaxial cable can realize a continuous and distributed deformation measurement, which is mainly used to measure underground shear deformation [24]. 

Here, a parallel helical sensing cable (PHSC), as a distributed sensor based on the TDR technique, is proposed to achieve the point positioning and deformation size measurement for a large-surface stretched deformation measurement. Compared with distributed optical fiber sensing technique, a PHSC can achieve precise point positioning of ground deformation; in addition, it can realize the transverse tensile measurement and enlarge the measurement range. In this study, we propose using a PHSC to realize the precise point positioning of the stretched deformation of the ground and the large deformation measurement of a landslide. First, the structure of a PHSC and the manufacturing process are introduced. Then, the relationship between characteristic impedance and stretched deformation is derived based on the proposed physical model. The principle of distributed deformation measurement based on TDR and PHSC characteristic impedance is also analyzed in detail. A laboratory calibration test and physical model test were carried out by the stretched test platform, and the procedure for these laboratory tests and the related results are presented here.

## 2. Mechanism of a Parallel Spiral Transmission Line

A PHSC used as a sensing component for the distributed measurement of ground deformation is typically buried under the ground. All the parameters of a PHSC, including the characteristic impedance and pitch, are basically fixed if there is no geological deformation. The characteristic impedance is changed when the PHSC is stretched, and the change of the characteristic impedance accurately reflects the stretched deformation. The characteristic impedance and the position of the deformation can be measured by the tensile test platform, the relationship between stretched deformation and the characteristic impedance can be found, and the determination position and the amount of stretched deformation of the surface and the underground deformation can also be determined [25]. The relationship between the characteristic impedance and the stretched deformation is the focus of this research. A photograph and structure diagram of the PHSC used here are shown in Figure 1. It is composed of a central silica gel strip, parallel copper wires of a silicone outer sheath, and an outer silicone protective sleeve [26]. The PHSC has good stretching ability, because the silica material has a certain elasticity and the parallel copper wires are spirally wound around the central silica gel strip. The length of the PHSC refers to the central silica gel strip length, and because they have the same structure, the same analysis principle is used in any parallel spiral transmission lines manufactured with different materials and sizes. The four types of PHSC are listed in Table 1, TS means that the conductors of the parallel wires are twisted, tinned copper wires. For example, the 24AWG PHSC is fabricated by 40 tinned copper wires with a diameter of 0.08 mm, the cross-sectional area of total 40 wires is 40 × π × 0.04^2^ mm^2^, and the equivalent diameter (*D*_e_) is about 0.5060 mm, obtained from the cross-sectional area.

## 3. Principle of the TDR Technique

The TDR technique is usually used for local ground deformation monitoring. TDR instruments are composed of a step signal generator with an 80E04 TDR module and a high-speed sampling oscilloscope of Tektronix DSA8300 (Tektronix, Shanghai, China). The measurement method is shown in Figure 2a. The beginning of the PHSC is connected to the 80E04 TDR module and the Tektronix DSA8300, and the end of the sensor is in an open state. The high-frequency step signal is transmitted into the PHSC, and the reflected signal is acquired by TDR instrument after a time delay. Then, the reflection coefficients and characteristic impedance of the sensing cable are calculated by sampling the reflected voltage signal. Figure 2b is a typical step signal incident and measured waveform, where *Z_DUT_* is the impedance of the tested devices, *Z*_0_ is the characteristic impedance of the TDR measurement system, which is 50 Ω, *V_incident_* is the incident step signal voltage value, *V_reflected_* is the reflected signal voltage value, and *V_measured_* is the measured voltage value, which is equal to (*V_incident_* + *V_reflected_*). 

The reflection coefficient *ρ* is written as follows:(1)$$\rho =\frac{V_{reflected}}{V_{incident}}=\frac{Z-Z_{0}}{Z+Z_{0}}$$

Therefore, the characteristic impedance *Z* of the PHSC can be obtained by the following [27,28]:(2)$$Z=Z_{0}\cdot \frac{1+\rho }{1-\rho }=Z_{0}\cdot \frac{V_{measured}}{2\cdot V_{incident}-V_{measured}}$$

When the PHSC is stretched, the stretched deformation becomes larger, which leads to a change in *Z*. The position of the characteristic impedance change is expressed as follows:(3)$$L_{\mathrm{location}}=\frac{v\cdot T}{2}$$
where the *L*_location_ is the reflected position from the beginning of the PHSC, *ν* is the transmission speed of the step signal in the PHSC, and *T* is the time difference between the reflection point and the input point measured by the TDR [29].

## 4. Distributed Parameters and Characteristic Impedance of PHSC

If the resistance of a 100 m length is less than 10 Ω for the four types of PHSC, then the influence of the distributed resistance can be neglected [30]. For lossless transmission lines or low-loss transmission lines, the characteristic impedance is expressed as
(4)$$Z\approx \sqrt{L/C}$$
where the *L* is the unit length conductance and *C* is unit length capacitance. The units of *L* and *C* are H/m and F/m, respectively.

In order to obtain the relationship between the distributed parameters and the stretched deformation (*s*) of the PHSC, the cross-section model of the PHSC in the stretched region is shown in Figure 3a. The PHSC is divided into the normal region, gradient region, and the stretched region. A_0_ and B_0_ are the two parallel copper wires in the center of the stretched region. There are (2*N* + 1) turns of spiral wires in the stretched region. *N* turns are in the left region, and another *N* turns are in the right region. The charge densities of two parallel copper wires A and B are −*η* and +*η*, respectively. The radius of the two wires are *r*, the fixed distance of the two parallel copper wires is *d*, and the pitch of spiral transmission line is (*d + s*). The diameter of the central silica gel strip is *D*, which is a structural constant, and the angle between the wire and the vertical direction is *θ*. Assuming that there exists a point P at the middle of B_0_ and A_R1_, based on the Biot–Savart law [31,32], the electric field strength *E* shown in Figure 3b along the *x*-axis direction at point P for a finite wire $B_{0}B_{0}^{\prime }$ with uniform current density is expressed as
(5)$$E_{B_{0}B_{0}^{\prime }}=2\int_{0}^{L_{p}}\frac{\eta }{4\pi \varepsilon }\cdot \frac{x}{{\left(x^{2}+y^{2}\right)}^{3/2}}dy=\frac{\eta }{2\pi \varepsilon }\cdot \frac{L_{p}}{x\sqrt{x^{2}+L_{P}{}^{2}}}$$
where the *ε* is relative permittivity and the *L*_p_ is the length of the wire $B_{0}B_{0}^{\prime }$ and Lp=D2cosθ. The *x* is the distance from point P to origin O.

According to the superposition theorem, the electric field strength sum at point P is given by
(6)$$E=E_{\mathrm{AL}}+E_{\mathrm{BL}}+E_{\mathrm{AR}}+E_{\mathrm{BR}}$$
where the first letter of the subscript represents the parallel wire A or B and the second letter indicates the parallel wire at the left or right of point P. 

Each electric field strength is expressed as
(7a)$$E_{\mathrm{AL}}=-\frac{\eta }{2\pi \varepsilon }\sum_{i=0}^{K}\frac{L_{p}}{[i(s+d)+d+x]\sqrt{{\left[i(s+d)+d+x\right]}^{2}+L_{p}{}^{2}}}$$
(7b)$$E_{\mathrm{BL}}=\frac{\eta }{2\pi \varepsilon }\sum_{i=0}^{K}\frac{L_{p}}{[i(s+d)+x]\sqrt{{\left[i(s+d)+x\right]}^{2}+L_{p}{}^{2}}}$$
(7c)$$E_{\mathrm{AR}}=\frac{\eta }{2\pi \varepsilon }\sum_{i=0}^{K}\frac{L_{p}}{[i(s+d)+s-x]\sqrt{{\left[i(s+d)+s-x\right]}^{2}+L_{p}{}^{2}}}$$
(7d)$$E_{\mathrm{BR}}=-\frac{\eta }{2\pi \varepsilon }\sum_{i=0}^{K}\frac{L_{p}}{[(i+1)(s+d)-x]\sqrt{{\left[(i+1)(s+d)-x\right]}^{2}+L_{p}{}^{2}}}$$
where the *K* is the number of PHSC turns on the left and right of point P.

Directed line segment *l* from the nearest point P along the B_0_ and A_R1_ direction is selected; if the direction of *l* is same as the electric field *E*, then the potential difference between B_0_ and A_R1_ is given by
(8)$$U=\int_{l}\vec{E}\cdot d\vec{l}=\int_{r}^{s-r}Edx$$

Equations (7a)–(7d) are brought into Equation (8), and the potential difference *U* can be derived as
(9)$$U=\frac{\eta }{2\pi \varepsilon }(-\zeta_{\mathrm{AL}}+\zeta_{\mathrm{BL}}+\zeta_{\mathrm{AR}}-\zeta_{\mathrm{BR}})$$
where *ζ*_AL_, *ζ*_BL_, *ζ*_AR_, and *ζ*_BR_ are the mathematical integral results. According to the characteristics of the definite integral and circuit symmetry, there are $\int_{r}^{s-r}E_{AL}dx=\int_{r}^{s-r}E_{BL}dx$ and $\int_{r}^{s-r}E_{AR}dx=\int_{r}^{s-r}E_{BL}dx$, which then lead to the following: (10a)$$\zeta_{\mathrm{AL}}=\zeta_{\mathrm{BR}}=\sum_{i=0}^{K}\ln\frac{\sqrt{{\left[(i+1)(s+d)-r\right]}^{2}+L_{p}{}^{2}}-L_{p}}{\sqrt{{\left[i(s+d)+(d+r)\right]}^{2}+L_{p}{}^{2}}-L_{p}}\frac{i(s+d)+d+r}{(i+1)(s+d)-r}$$
(10b)$$\zeta_{\mathrm{AR}}=\zeta_{\mathrm{BL}}=\sum_{i=0}^{3}\ln\frac{\sqrt{{\left[i(s+d)+s-r\right]}^{2}+L_{p}{}^{2}}-L_{p}}{\sqrt{{\left[i(s+d)+r\right]}^{2}+L_{p}{}^{2}}-L_{p}}\frac{i(s+d)+r}{i(s+d)+s-r}$$

In order to simplify the calculation and guarantee the calculation precision, set *K* = 3. Then, for 22AWG PHSC (*r* = 0.3099 mm, *d* = 1.6 mm, *D* = 4.6 mm, *θ* = 30°), the initial value of *s* is equal to *d,* the curves of *ζ*_AL_, *ζ*_BL_ and (*ζ*_BL_
*− ζ*_AL_) with *s* are shown in Figure 4. When the PHSC is stretched, the *s* becomes larger, and all the results increase, but the rate of increase slows down.

Based on the Gauss theorem, it is known that the capacitance per unit length between B_0_ and A_R1_ is the ratio of the charge density to the voltage, which is given by
(11)$$C=\frac{\eta }{U}$$

According to Equation (11), the electric field strength becomes larger with the increase of *s*, but the charge density is a constant. Additionally, the distribution capacitance becomes smaller and the distribution inductance becomes larger. Then, the distributed capacitance *C* is obtained as
(12)C=πε(ζBL−ζAL)

The constitutive parameters (*μ*,*ε*) in the medium are given by
(13)LC=με

Then, the distributed inductance can be expressed as
(14)$$L=\frac{\mu }{\pi }\cdot (\zeta_{\mathrm{BL}}-\zeta_{\mathrm{AL}})$$
where the *μ* is relative permeability. 

Based on Equations (4), (12), and (14), the characteristic impedance *Z* of PHSC is obtained as [33]
(15)$$Z=\sqrt{\frac{\mu }{\varepsilon }}\cdot \frac{\zeta_{\mathrm{BL}}-\zeta_{\mathrm{AL}}}{\pi }$$

It can be seen from Equation (15) that *Z* is only related to the physical size of *r*, *d*, *s*, *D*, and *θ*. The relative dielectric constant ε is 5.5, and the relative permeability *μ* is 1. The distribution capacitance *C* and inductance *L* of 22AWG PHSC (*r* = 0.3099 mm, *d* = 1.6 mm, *D* = 4.6 mm, *θ* = 30°) with the *s* are shown in Figure 5. It can be observed that the distribution capacitance *C* becomes smaller with the increase of *s*, but the distribution inductance *L* becomes larger. 

The characteristic impedance *Z* of four types of PHSC are shown in Figure 6. It can be seen that Z decreases as the value of *D*_e_ increases. When the PHSC is stretched, *s* becomes larger. *Z* gradually increases with the increase of *s*, but the rate of increase slows down. Z at the center point of the PHSC is a maximum when *s* is a fixed value. Because the deformation region is divided into the stretched deformation region, gradual region, and a normal region, *Z* will be decreased from the maximum at the center point of the stretched region to the minimum of the normal region.

In application, the tensile deformation should be obtained based on the measured characteristic impedance of the PHSC. However, considering that Equation (15) is very complex, the inverse function is not easy to obtain. A simple inverse function is easily obtained by curve fitting based on the theoretically calculated results shown in Figure 6. The inverse function for four types PHSC is given by
(16)$$s=\frac{a}{\ln\left(\frac{Z+b}{c}\right)}$$
where *a*, *b*, and *c* are the fitting coefficients.

## 5. Measurement Experiments

### 5.1. Tensile Test Platform

In order to verify the theoretical derivation correctness and realize the distributed deformation measurement of the ground, it is necessary to establish a PHSC tensile test platform and carry out a stretching experiment. The established tensile test platform shown in Figure 2 is composed of the stretching structure, a TDR measurement system, and a host computer [34]. The stretched structure shown in Figure 7 is mainly composed of mechanical structure and a stepper motor. 

The software interface for the distributed measurement of deformation based on the TDR technique is shown in Figure 8, and it consists of a waveform display, parameter setting, data sampling, data analysis, and result export. The waveform display function can display the time–domain reflection waveforms of multiple results in one picture to facilitate the observation of the waveform change of the deformation position and also actively identify and mark the starting point and endpoint of the PHSC. The unit of the ordinate axis is set to the characteristic impedance. The length of the PHSC and the parameters of function by curve fitting based on the theoretically calculated results are written into the parameter setting region. The sampling command is sent to the TDR by the data sampling function. Then, the waveforms are acquired and sent to the host computer. The data analysis function is used to analyze the acquired waveforms. First, according to the waveforms, the beginning and the end of the PHSC are located, and the waveform propagation time is calculated. The transmission speed of the step signal in the PHSC is obtained under the length of the PHSC, which is known. Second, the stretched position and the stretch region are found based on the change of the characteristic impedance. The stretched deformation *s* is calculated according to Equation (16). The amount of the stretched deformation is a product of (*s*–*d)*, and the parallel wire turns in the stretched region. For many stretched regions, the positions can be found, and the acquired waveforms are analyzed, respectively. 

### 5.2. Point Positioning Calibration Experiment

To determine the distributed measurement of the ground stretched deformation by using PHSC, the first step is to locate the position of the stretched region. In many cases, multiple positions are simultaneously stretched, while ground deformations occur in many places. Therefore, multiple stretched points should be located and calculated. A 6.6 m AWG22 PHSC (*r* = 0.3099 mm, *d* = 1.6 mm, *D* = 4.6 mm, *θ* = 30°, *ε* = 5.5) was selected for the stretched deformation experiment, and the stretching center point was selected to be 2, 3.5, and 5 m from the beginning of the PHSC, respectively. The tensile region was set to 140 mm, and the amount of the deformation was set to 20 mm. A photograph of the multi-point positioning measurement experiment is shown in Figure 9. A set of measured results are listed in Table 2. It can be seen that the measurement of the absolute errors was less than 0.01 m, and the results have an accuracy rating of 0.2.

### 5.3. Stretched Deformation Experiment

A 6.6 m AWG24 PHSC (*r* = 0.25303 mm, *d* = 1.6 mm, *D* = 4.6 mm, *θ* = 30°, *ε* = 5.5) was selected for the tensile deformation experiment, and the stretching center point was selected to be 2 and 4 m from the start point of the test PHSC, respectively. The stretched region was set to 200 mm, the initial value of *s* was 1.6 mm, and there were 62.5 turns in the tensile region. The maximum value of the characteristic impedance was measured by the PHSC tensile test platform. A set of measured results were selected and are listed in Table 3. The comparison between the measured results and the theoretically calculated results based on the distributed parameters model is shown in Figure 10. 

The theoretically calculated results are in good agreement with those measured by the experiment, verifying the reliability of the physical model and the measurement method. The measured results at 2 m from the beginning of the PHSC are in good agreement with the theoretically calculated results, which is mainly due to the end of the test PHSC in the open state. The incident signal is a total reflection station at the end of the PHSC. The reflected signal voltage near the end changed relatively drastically, and the characteristic impedance changed relatively drastically, which affected the measurement accuracy.

The inverse function can be obtained by curve fitting based on the theoretically calculated results shown in Figure 6 for AWG24 PHSC. The coefficients of Equation (16) were filled into the host computer’s software interface, as shown in Figure 8. The local stretched deformation measurement was carried out for the same 6.6 m AWG24 PHSC (*r* = 0.2530 mm, *d* = 1.6 mm, *D* = 4.6 mm, *θ* = 30°, *ε* = 5.5), and the stretching center point was selected to be a position 2.5 and 4.5 m from the starting end of the PHSC, respectively. The stretched region was set to 200 mm, the initial value of *s* was 1.6 mm, and the turns of the PHSC in the tensile region were 200/3.2, which is 62.5. The amount of the stretched deformation was increased from 0 to 50 mm with an initial step with a 5 mm increase. The incremental step of *s* is 0.080 mm from the initial value. The sampling characteristic impedance curves cut from the host computer software interface are shown in Figure 11. The measured characteristic impedance at the center point of the stretched region is a maximum, and there is a decrease from the maximum to the minimum in the normal region. The formation mechanism of the characteristic impedances in the partial amplified area is consistent with the previous theoretical analysis. The measured characteristic impedances and tensile amount are listed in Table 4. When the amount of stretched deformation is small, the absolute errors are within 2 mm. As the amount of the stretched deformation increases, the absolute error increases and is less than 3 mm, from the 0 to 50 mm stretching amount. 

## 6. Conclusions

This research describes the mechanism of using a PHSC for the distributed measurement of ground deformation. The following conclusions have been drawn based on the theoretical analysis and experimental results.(1)Ground deformation monitoring based on the TDR technique and proposed PHSC is a very feasible technique, and the sensing parallel spiral transmission line implantation method is quite effective.(2)A distribution parameter calculation model of PHSC is proposed and used to derive the distributed capacitance, distributed inductance, and characteristic impedance. The distribution capacitance becomes smaller with the increase of stretched s, and distribution inductance becomes larger. The relationship between the stretched deformation and the characteristic impedance of PHSC can be found based on the model and is shown here as Equation (15). It can be seen from Equation (15) that the impedance increases when the cable is stretched.(3)The derived characteristic impedance is related to the physical size of the equivalent radius of the two wires, the fixed distance between the two parallel copper wires, the pitch of the spiral transmission line, the diameter of the central silica gel strip, and the angle between the wire and the vertical direction. The characteristic impedance decreases with the value of the equivalent radius of the two wires.(4)The characteristic impedance at the stretched center point is a maximum when s is a fixed value. The deformation region is divided into the stretched deformation region, the gradual region, and a normal region. The characteristic impedance decreases from the maximum at the center point of stretched region to the minimum of the normal region.(5)In application, tensile deformation can be obtained based on the measured characteristic impedance of the PHSC. The function of the stretched deformation and characteristic impedance can be obtained by curve fitting based on the theoretically calculated results. According to the function, the stretched deformation measurement is carried out.(6)A measurement experiment platform, including a stretching device, TDR measurement system, and host computer software, was established to carry out the multi-point positioning measurement and the amount of stretched deformation measurement at different positions. The theoretically calculated results are in good agreement with the experiment results, which show that the PHSC can measure the rock–soil large deformation and verify the accuracy of the theoretical derivation.

In future work, the model for the characteristic impedance and stretched deformation should be further optimized to improve the measurement accuracy. The parallel helical sensing cable should be applied to monitor the distributed measurement of ground deformation.

## Figures and Tables

**Figure 1 sensors-19-01297-f001:**
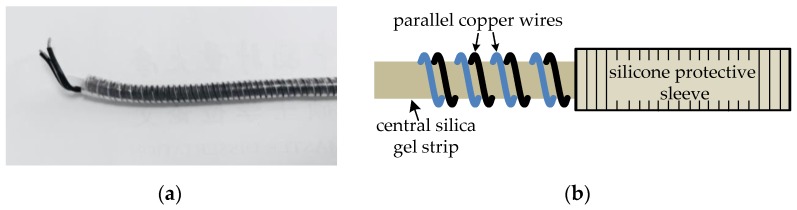
(**a**) Photograph and (**b**) structure diagram of a parallel helical sensing cable (PHSC).

**Figure 2 sensors-19-01297-f002:**
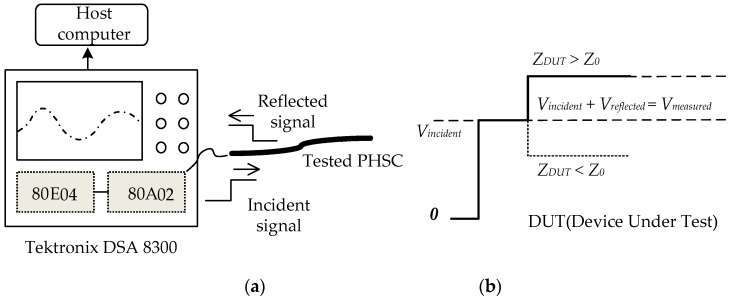
Diagram of tensile test platform and waveform (**a**) tensile test platform based on the time–domain reflectometry (TDR) technique; (**b**) a typical step signal incident and measured waveform.

**Figure 3 sensors-19-01297-f003:**
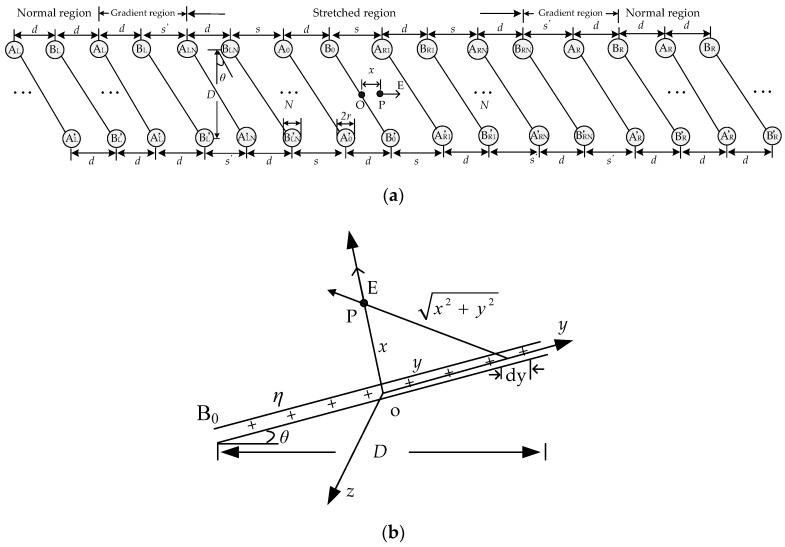
The cross-section model of the PHSC for the distributed parameters (**a**) cross-section model; (**b**) diagram of the electric field distribution.

**Figure 4 sensors-19-01297-f004:**
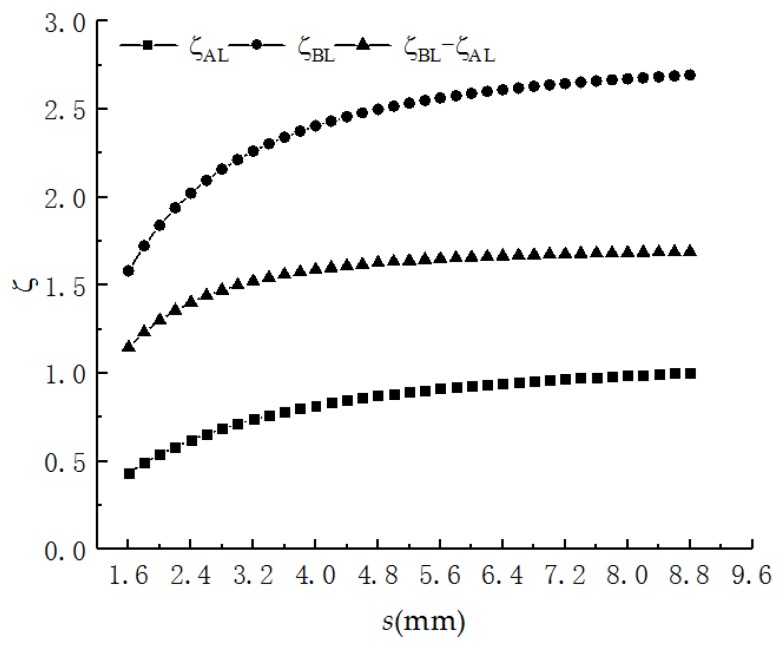
The theoretically calculated results of *ζ*_AL_, *ζ*_BL_ and (*ζ*_BL_
*− ζ*_AL_).

**Figure 5 sensors-19-01297-f005:**
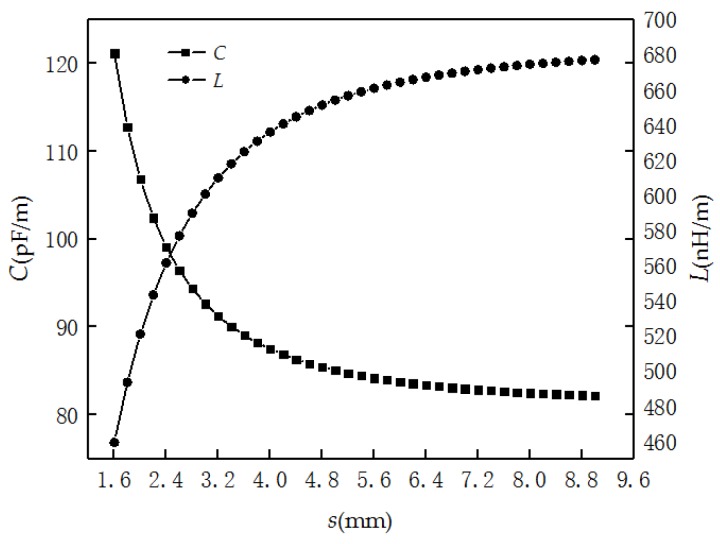
The calculated distribution capacitance *C* and inductance *L* of 22AWG PHSC.

**Figure 6 sensors-19-01297-f006:**
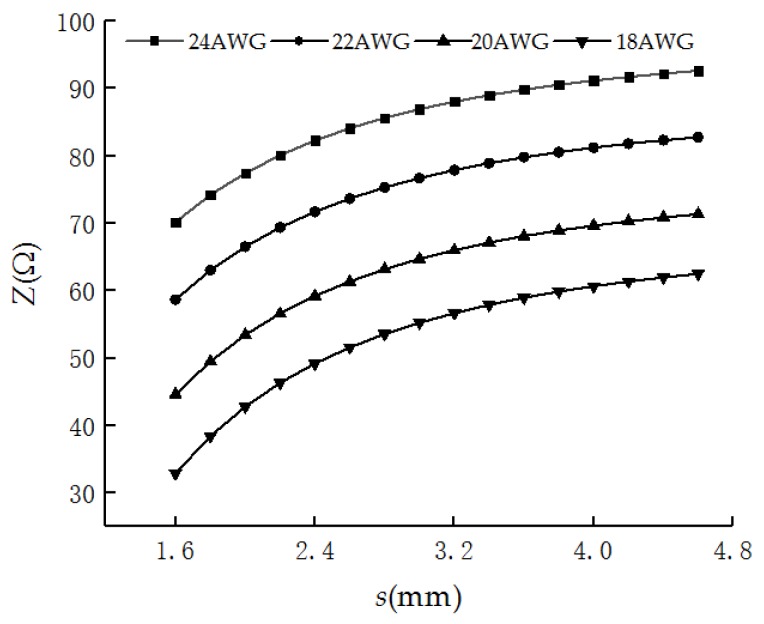
The calculated characteristic impedance *Z* of the four types of PHSC.

**Figure 7 sensors-19-01297-f007:**
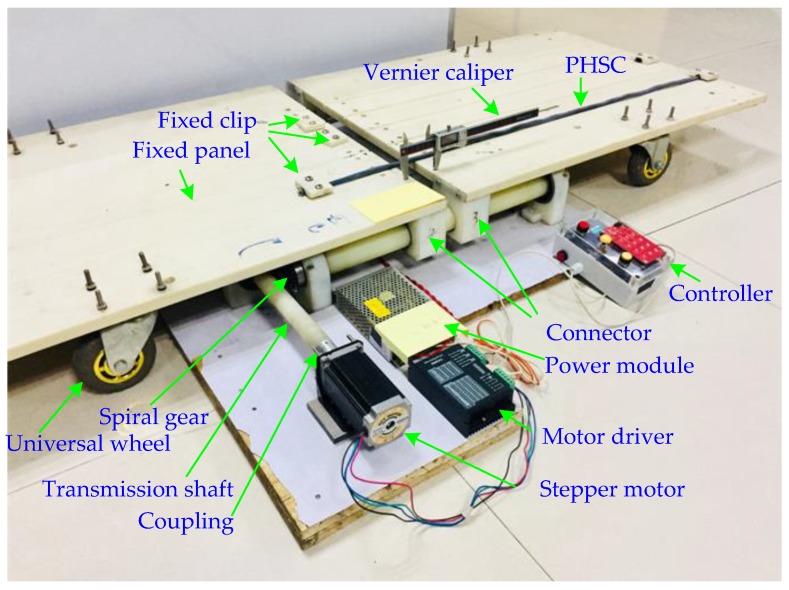
The PHSC tensile test platform.

**Figure 8 sensors-19-01297-f008:**
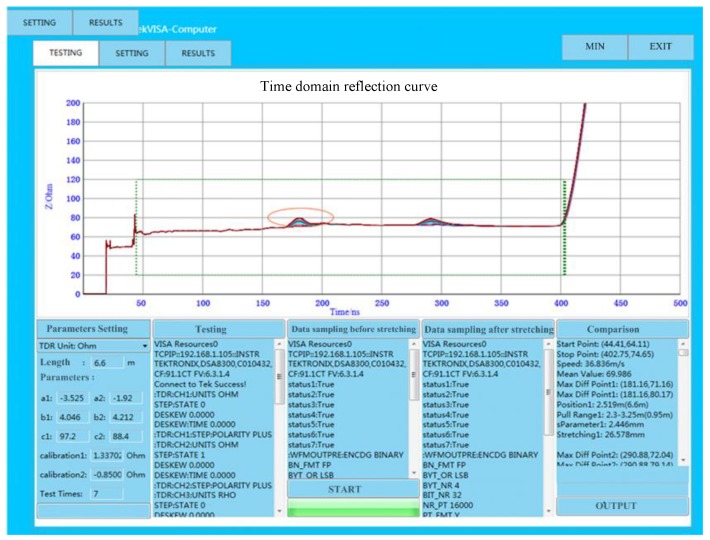
Software interface for the distributed measurement of deformation based on the TDR technique.

**Figure 9 sensors-19-01297-f009:**
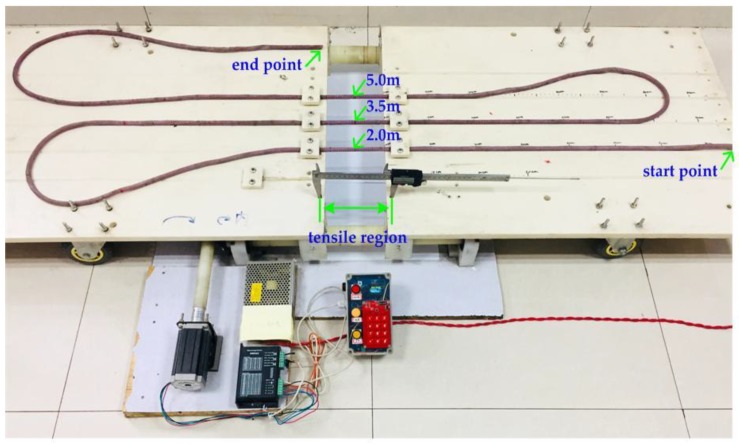
A photograph of the multi-point positioning measurement experiment.

**Figure 10 sensors-19-01297-f010:**
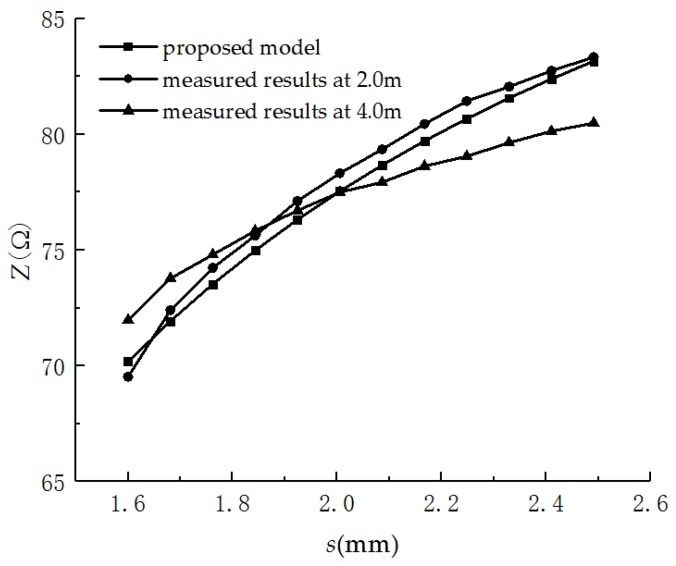
The comparison between the measured results and the theoretically calculated results.

**Figure 11 sensors-19-01297-f011:**
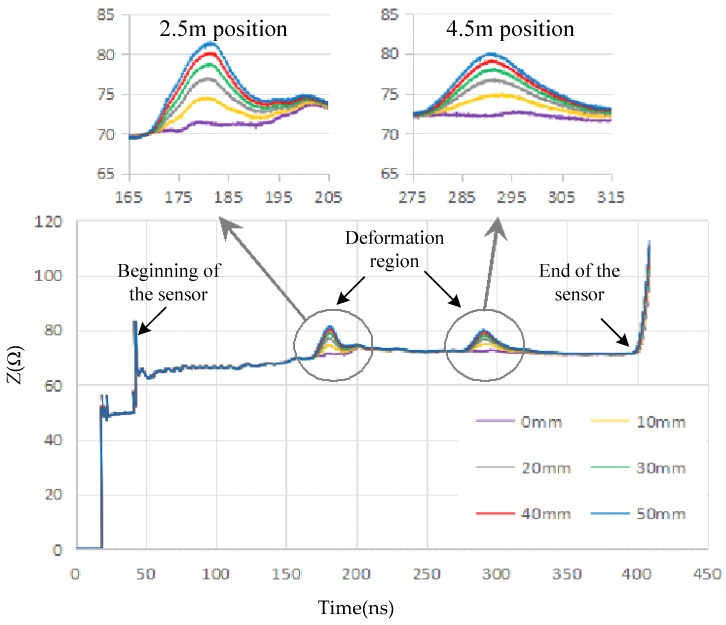
Sampling characteristic impedance curves.

**Table 1 sensors-19-01297-t001:** Four types of PHSC.

Types	Parameters	*D*_e_ (mm)
24AWG	40 × 0.08TS	0.5060
22AWG	60 × 0.08TS	0.6197
20AWG	100 × 0.08TS	0.8000
18AWG	150 × 0.08TS	0.9798

**Table 2 sensors-19-01297-t002:** Multi-point positioning measurement results.

Position(m)	Measured Result(m)	Absolute Error(m)	Reference Error
2.000	1.993	−0.007	−0.106%
3.500	3.501	0.001	0.015%
5.000	5.006	0.006	0.091%

**Table 3 sensors-19-01297-t003:** The *s* and measured maximum *Z* at the 20 and 4 m positions.

*s* (mm)	*Z*_2m_ (Ω)	*Z*_4m_ (Ω)	*s* (mm)	*Z*_2m_ (Ω)	*Z*_4m_ (Ω)
1.600	69.53	71.98	2.084	79.35	77.93
1.681	72.41	73.78	2.165	80.45	78.63
1.761	74.23	74.80	2.245	81.44	79.05
1.842	75.62	75.83	2.326	82.06	79.64
1.923	77.12	76.71	2.406	82.75	80.14
2.003	78.32	77.50	2.487	83.34	80.49

**Table 4 sensors-19-01297-t004:** The measured results at the 2.5 and 4.5 m positions.

Set Tensile Amount (mm)	Tensile Position (m)	*s*(mm)	*Z*(Ω)	Measured Tensile Amount (mm)	Absolute Error (mm)
5.000	2.500	1.680	72.96	4.879	−0.121
	4.500		73.79	3.944	−1.056
10.000	2.500	1.760	74.52	9.601	−0.399
	4.500		74.83	9.081	−0.919
15.000	2.500	1.840	75.81	14.112	−0.888
	4.500		75.89	15.806	0.806
20.000	2.500	1.920	76.96	18.959	−1.041
	4.500		76.76	21.221	1.221
25.000	2.500	2.000	77.77	23.593	−1.407
	4.500		77.46	26.147	1.147
30.000	2.500	2.080	78.67	28.785	−1.215
	4.500		78.21	31.470	1.470
35.000	2.500	2.160	79.46	33.566	−1.434
	4.500		78.57	36.747	1.747
40.000	2.500	2.240	80.17	38.478	−1.522
	4.500		79.14	41.964	1.964
45.000	2.500	2.320	80.94	43.196	−1.804
	4.500		79.66	47.415	2.415
50.000	2.500	2.400	81.37	47.740	−2.260
	4.500		80.05	52.820	2.820

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
