# Peer review of "Using a Parallel Helical Sensing Cable for the Distributed Measurement of Ground Deformation"

_sensors, 2019, doi:10.3390/s19061297_

Reviewer 1 Report

The introduction can clearly present the background of the topic, the progress and classifications of previous methods, the gaps in the current research and the frame of the article. In terms of research gaps summarised in the introductory section, the research problem is clearly outlined. The research problem is an innovative study 

The data processing and the selection phase of the variables are scientifically valid. The data used in the study shall be clearly presented. The results shall be presented in a comprehensive manner using high quality figures and maps. The results are presented and explained with detailed and correct statements.

In the conclusion section, the phenomenon reflected by the results are further discussed and can reasonably respond to the research problem, and in this study are supported by the relevant reference results. Future recommendations are constructive for future research based on this study.

Author Response

Dear reviewer:

Thank you very much for your comments on our manuscript entitled “Study on the Mechanism of Parallel Helical Sensing Cable Used for Ground Deformation Distributed Measurement”. The comments listed by reviewer are all very valuable and helpful for revising and improving our paper, as well as the important guiding significance to our research. We have studied the comments carefully and have modified the manuscript which we hope to meet with approval (marked in high light). The responses to the reviewer’s comments are as following:

Comment 1: The introduction can clearly present the background of the topic, the progress and classifications of previous methods, the gaps in the current research and the frame of the article. In terms of research gaps summarised in the introductory section, the research problem is clearly outlined. The research problem is an innovative study.  

Response: According to your suggestions, the progress and classifications of previous methods has been reorganized and written in introduction (line 35-39). The gaps in the current research has been discussed in detail from line 40 to 69. The research problem is clearly outlined from line 70 to 74. The corresponding references have been adjusted and added .The research problem is clearly outlined:A PHSC as a distributed sensor based on the TDR technique is proposed to achieve the point positioning and the deformation size measurement for the surface large stretched deformation measurement.

Comment 2: The data processing and the selection phase of the variables are scientifically valid. The data used in the study shall be clearly presented. The results shall be presented in a comprehensive manner using high quality figures and maps. The results are presented and explained with detailed and correct statements.

Response:

According to your suggestions, the equations 10(a) and 10(b) are completely deduced and given the final result (line 172-173).

Based on the Gauss theorem, the equation (11) is simplified( line 10-11), the result of the equation (11) is discussed from line 181 to 183.

The amount of the stretched deformation is modified in the line 234. The accuracy of calculation and measurement data is unified and detected, leaving four digits after the decimal point.

For Figure 10, the measured results at 2.000m from the beginning of PHSC are in good agreement with the theoretical calculation results, and the reasons for the measured accuracy is better than the results at 4.000m position have been analyzed(line 263-268).

For Figure 11, the measurement process has been further illustrated (line 272-273, line 277-280), and for the Table 4, the column of stretch deformation was added and the title was rectified.

The grammar throughout the paper has been further checked.

Comment 3:In the conclusion section, the phenomenon reflected by the results are further discussed and can reasonably respond to the research problem, and in this study are supported by the relevant reference results. Future recommendations are constructive for future research based on this study.

Response: According to your suggestions, the conclusion section has been modified (line 306-316). The future research based on this study has been added after the conclusion (line 326-328).

Thank you again for your comments, and we hope our responses and corrections would meet with your approval

Reviewer 2 Report

Please check the grammar throughout the paper.

I recommend to strengthen the introduction

I recommed to better specify or explain the result or discussion of the expriments

I recommend to strengthen the conclusion

Author Response

Dear reviewer :

Thank you very much for your comments on our manuscript entitled “Study on the Mechanism of Parallel Helical Sensing Cable Used for Ground Deformation Distributed Measurement”. The comments listed by reviewer are all very valuable and helpful for revising and improving our paper, as well as the important guiding significance to our research. We have studied the comments carefully and have modified the manuscript which we hope to meet with approval (marked in high light). The responses to the reviewer’s comments are as following:

 Comment 1:  Please check the grammar throughout the paper

 Response: According to your suggestions, The grammar throughout the paper has been further checked.

    The accuracy of calculation and measurement data is unified and detected, leaving four digits after the decimal point.

    The measurement absolute errors of position are less than 0.01m based on the Table 2, not 0.01mm, the value was rectified in line 24 and line 246. The unit of absolute in Table 2 was rectified.

    In line 165, "The equations of (7a)-(74d)......" was modified to " The equations of (7a)-(7d)......".

    In line 167,  "where all the ζ are the mathematical integral results." was modified to " where ζAL, ζBL, ζAR and ζBR are the mathematical integral results". In line 173,  "all the ζ..." was modified to" all the results...".

Comment 2: I recommend to better specify or explain the result or discussion of the experiments

Response: According to your suggestions, the equations 10(a) and 10(b) are completely deduced and given the final result (line 172-173).

Based on the Gauss theorem, the equation (11) is simplified( line 10-11), the result of the equation (11) is discussed from line 181 to 183.

The amount of the stretched deformation is modified in the line 234.

For Figure 10, the measured results at 2.000m from the beginning of PHSC are in good agreement with the theoretical calculation results, and the reasons for the measured accuracy is better than the results at 4.000m position have been analyzed(line 263-268).

For Figure 11, the measurement process has been further illustrated (line 272-273, line 277-280), and for the Table 4, the column of stretch deformation was added and the title was rectified.

Comment 3: I recommend to strengthen the introduction

Response: According to your suggestions, the progress and classifications of previous methods has been reorganized and written in introduction (line 35-39). The gaps in the current research has been discussed in detail from line 40 to 69. The research problem is clearly outlined from line 70 to 74. The corresponding references have been adjusted and added .

Comment 4: I recommend to strengthen the conclusion

 Response: According to your suggestions, the conclusion section has been modified (line 306-316).

Thank you again for your comments, and we hope our responses and corrections would meet with your approval

Reviewer 3 Report

In an abstract, the authors pointed out that the article proposed a technique that was used to monitor large ground deformations. The key is to know what it means "large ground deformations" ? What are the units ? 

In this way, I expected detailed descriptions of technology (for the possibility of its reproduction) together with a practical experiment in the field. Reading the article I found only a laboratory test, which in my opinion is insufficient. This gives information only about the review of the method, where the use of which can be placed under a big question mark.There is no information on how effective this method is? With what sensors can you perform a data fusion ? What is the applicability of the solution etc.

From my perspective, GPR (Ground Penetrating Radar) gives much better results as well as geotechnical wells. They are commonly used to determine the laying of soil layers, however, in the introduction the authors omitted to apply these solutions. 

An example of a measurement publication using a scanner and GPR is "The application of non-destructive methods in the diagnostics of the approach pavement at the bridges" 

Author Response

Dear reviewer :

Thank you very much for your comments on our manuscript entitled “Study on the Mechanism of Parallel Helical Sensing Cable Used for Ground Deformation Distributed Measurement”. The comments listed by reviewer are all very valuable and helpful for revising and improving our paper, as well as the important guiding significance to our research. We have studied the comments carefully and have modified the manuscript which we hope to meet with approval (marked in high light). The responses to the reviewer’s comments are as following:

Comment 1: In an abstract, the authors pointed out that the article proposed a technique that was used to monitor large ground deformations. The key is to know what it means "large ground deformations" ? What are the units ?

Response: The parallel helical sensing cable based on the time-domain reflectometry (TDR) technique is proposed and used to monitor the large ground stretched deformation, which is compared with distributed optical fiber sensing technique, when the amount of deformation for the rock-soil is large than 2mm, the optical fiber is easy to be broken due to the limitation of stretching ability. This was added in the introduction (line 62-64). The PHSC has the large stretch ability because the silica material has a certain elasticity and the parallel copper wires are spirally wound around the center silica gel strip.

   The fixed center distance is d, and the pitch is d+s. The initial value of s is equal to d and becomes larger with the extension according to the parallel helical sensing cable internal structure, The initial value of s is equal to 1.6mm in manufacturing. Silicone material was selected high temperature vulcanized silicone rubber, elongation at break is greater than 300% based on the experiments. It takes five steps to manufacture the parallel helical sensing cable. The first step shown in Figure 1(a) is to make the center silica gel strip of the parallel helical sensing cable. In step 2 shown in Figure 1 (b), the two copper wires wrapped together by the silica gel are twisted around the winding wheel. In step 3 shown in Figure 1 (c), the parallel copper wires on the winding wheel are helically twisted around the center silica gel strip. In step 4 shown in Figure 1 (d), the outer layer is wrapped with silica gel. The last step shown in Figure 1 (e) is to undergo the high temperature vulcanization. The photos come from local joint laboratory for geological hazards research and monitoring of China Jiliang university, Hangzhou.

Figure 1 Manufacturing process and physical photograph of parallel helical sensing cable

Comment 2:In this way, I expected detailed descriptions of technology (for the possibility of its reproduction) together with a practical experiment in the field. Reading the article I found only a laboratory test, which in my opinion is insufficient. This gives information only about the review of the method, where the use of which can be placed under a big question mark. There is no information on how effective this method is? With what sensors can you perform a data fusion ? What is the applicability of the solution etc.

Response:  The parallel helical sensing cable is designed and used to monitor the ground deformation , processing technology research has been successfully completed, the corresponding technology has applied for a USA national patent and is authorized (Sensing cable with parallel spiral transmisson line structure for distributed sensing and measuring of rock-soil mass deformation, USA,US 9,618,644 B2), the main purpose of this paper is study on the mechanism of parallel helical sensing cable for ground deformation distributed measurement. The principle of distributed deformation measurement based on TDR technique and PHSC characteristic impedance is analyzed in detail.    

      The laboratory calibration test is carried out by the designed tensile test platform. Experiments are divided into multiple point positioning measurement experiment, stretched measurement experiment at 2.0m, 2.5m, 4.0m and 4.5m, all experiments are repeated multiple times, the selected experimental measurement results are listed in Table 2, Table 3 and Table 4. The measured results are in good agreement with the theoretical calculation results.

  According to your suggestions, The future research based on this study has been added after the conclusion (line 326-328).

Comment 3: From my perspective, GPR (Ground Penetrating Radar) gives much better results as well as geotechnical wells. They are commonly used to determine the laying of soil layers, however, in the introduction the authors omitted to apply these solutions. 

An example of a measurement publication using a scanner and GPR is "The application of non-destructive methods in the diagnostics of the approach pavement at the bridges" 

Response:  The working principle of GPR (Ground Penetrating Radar) is to infer the spatial position, structure, shape and burial depth of the underground medium based on the characteristics of the received electromagnetic wave waveform, amplitude and time. The GPR technique is widely used in depth and thickness of different rock layers detection, dam body leakage detection, underground metal or non-metallic pipes, cables, holes, foundation layers, steel bars in concrete and other underground buried parts. A parallel helical sensing cable as a distributed sensor based on the TDR technique is proposed and designed in the paper to achieve the point positioning and the deformation amount for the ground stretched deformation distributed measurement. The parallel helical sensing cable is used to monitor rock-soil and ground stretched deformation which occurs in the horizontal direction

According to your suggestions, the reference of  "Miskiewicz, M.; Lachowicz, J.; Tysiac, P.; Jaskula, P.; Wilde, K. The application of non-destructive  methods in the diagnostics of the approach pavement atthe

 bridges. Conference on Resilient and Safe Road Infrastructure, Kielce, Poland, May 08-09, 2018, 36."  has been cited in the paper, the reference number is 22, the main work of this paper was also briefly described from line 56 to 58 in the introduction.

Thank you again for your comments, and we hope our responses and corrections would meet with your approval.

Round  2
